# An evaluation of clouds and radiation in a Large-Scale Atmospheric Model using a Cloud Vertical Structure classification

by

Dongmin Lee[1,2], Lazaros Oreopoulos[2], and Nayeong Cho[3,2]

*1. Morgan State University*

*2. NASA Goddard Space Flight Center*

*3. University Space Research Association*

Confidential manuscript revised for the

*Geoscientific Model Development*

**Corresponding author address:**

*Dongmin Lee*

*NASA-GSFC*

*Code 613*

*Greenbelt MD 20771*

*Dongmin.Lee@nasa.gov*

**Abstract**
We revisit the concept of Cloud Vertical Structure (CVS) classes we have previously employed to
classify the planet's cloudiness (Oreopoulos et al., 2017). The CVS classification reflects simple
combinations of simultaneous cloud occurrence in the three standard layers traditionally used
to separate low, middle, and high clouds and was applied to a dataset derived from active lidar
and cloud radar observations. This classification is now introduced in an Atmospheric Global
Climate Model, specifically a version of NASA's GEOS-5, in order to evaluate the realism of its
cloudiness and of the radiative effects associated with the various CVS classes. Such classes can
be defined in GEOS-5 thanks to a subcolumn cloud generator paired with the model's radiative
transfer algorithm, and their associated radiative effects can be evaluated against observations.
We find that the model produces 50% more clear skies than observations in relative terms, and
produces isolated high clouds that are slightly less frequent than in observations, but optically
thicker, yielding excessive planetary and surface cooling. Low clouds are also brighter than in
observations, but underestimates of the frequency of occurrence (by ~20% in relative terms)
help restore radiative agreement with observations. Overall the model reproduces better the
longwave radiative effects of the various CVS classes because cloud vertical location is
substantially constrained in the CVS framework.

## 1. Introduction

The large impact of clouds on the Earth's radiation budget and the growing wealth of satellite-based cloud observations are strong motivators for their systematic assessment in climate models. Such evaluation exercises focus on either cloud properties, or metrics of cloud radiative impact, or ideally on both (Pincus et al., 2008; Nam et al., 2012; Klein et al., 2013; Wang and Su, 2013; Dolinar et al., 2015).

Assessments of cloud properties with satellite observations are not always straightforward for a variety of reasons such as inability to define in the model a particular satellite-observed property, or limitations in the satellite observations. For example, the vertically integrated cloud optical depth of the cloudy portion of a model grid cell is an ill-defined quantity that cannot be obtained trivially from the model's optical depth profile since it is intimately associated with a cloud fraction profile, making thus layer optical depths relevant for only the cloudy portions of the grid cell which vary by model height and can conceptually be vertically aligned in various ways. In contrast, vertically-integrated cloud optical depth is quite robustly defined in observations since it is measured with passive imagers at a much higher resolution for which overcast conditions can be more safely assumed. Issues such as these have led to the development of "satellite simulators" that transform Global Climate Model (GCM) cloud fields to forms that are closer analogues to their counterparts observed by satellites (e.g., the COSP simulator – Bodas-Salcedo et al., 2011).

The quality of simulated clouds in GCMs can also be measured in terms of the realism of their radiative impact using quantities such as the Cloud Radiative Effect (CRE), i.e., the difference between all-sky and clear-sky fluxes at the spatial scales of a model grid cell (Wang and Su, 2013). This type of comparison can be performed at a variety of spatiotemporal scales and is often quite illuminating, but the interpretation of findings can suffer from inconsistencies in how the estimates are obtained for satellites and models.

This paper is yet another attempt to evaluate clouds in an atmospheric GCM (AGCM), specifically a version of the Goddard Earth Observing System version 5 (GEOS-5) model (Rienecker et al., 2008; Molod et al., 2012), a multi-purpose global model that is used for a variety of applications. Both approaches of cloud assessment are used, namely comparison of

the cloud fields themselves, but also comparison of cloud radiative impacts. Our cloud property
evaluation focuses on a single aspect of cloudiness: Cloud Vertical Structure (CVS). The
comparison is possible because of recent progress in two areas: active cloud remote sensing
which makes resolving cloud vertical profiles possible; and the development of schemes
(subcolumn generators) that create subgrid cloud vertical structures in GCMs. Being able to
categorize clouds in terms of a few CVS categories facilitates the comparison between
observations and models and enables a more rigorous CRE comparison that evaluates the
model's skill with regard to how it simulates the radiative impact of individual CVS classes.
**2. Data and methodology**
The observational reference dataset of CVS class occurrence and associated radiative fluxes is
essentially the same as Oreopoulos et al. (2017), hereafter O17, and spans four years (2007-
2010.) A schematic illustration of the original CVS classes of O17 is reproduced here as Fig. 1.
The details of how cloud layer boundaries available in the 2B-CLDCLASS-LIDAR R04 dataset
(Sassen and Wang, 2012, see also http://tinyurl.com/2b-cldclass-lidar), a joint product coming
from CloudSat and CALIPSO (hereafter, "CC") active cloud radar and lidar observations, were
interpreted as cloud layer profiles belonging to one of these classes are described exhaustively
in the appendix of O17. The definition of the CVS classes hinges on defining broad categories of
high $H$, middle $M$, and low $L$ clouds which are confined to three standard atmospheric layers,
one above 440 hPa, another between 680 and 440 hPa and yet another below 680 hPa,
respectively. The vertical level boundaries defining these standard layers come from the
International Satellite Cloud Climatology Project (ISCCP), (Rossow and Schiffer, 1991). The
reference radiative fluxes come from the 2B-FLXHR-LIDAR R04 CC product (L'Ecuyer et al., 2008;
Henderson et al., 2013; Matus and L'Ecuyer, 2017) and are obtained from a radiative transfer
algorithm operating on observed and re-analysis output which has at its core retrieved CC cloud
properties.

27       For the purposes of this study, the CVS classes have been reduced to seven by merging

the CVS classes for which clouds occur simultaneously in the same two or three standard
adjacent layers (all multi-layer CVS classes other than "HL"). In other words, we do not
distinguish any longer between CVS classes with clouds occurring in the same adjacent standard
layers, even if those were previously discerned based on whether or not a clear layer of
substantial vertical extent was present to separate the cloud layers. This means in practice that
we do no longer distinguish (cf. Fig. 1) between CVS = "H×M×L" and "HML" (now simply
"HML"), CVS = "H×M" and "HM" (now simply "HM"), CVS = "M×L" and "ML" (now simply "ML").
The reason for reducing the CVS classes to seven from the original ten is the complexity of the
model cloud profiles which can consist of numerous distinct cloud layers and which therefore
renders the O17 CVS classification scheme inapplicable. The original scheme was designed for
observed cloud profiles from CC that rarely (less than 1% of the time) consisted of more than
four distinct cloud layers in which case they were either ignored or processed only in the
simplest of cases (such as multiple individual cloud layers residing within a single standard layer
– see appendix of O17).

12        A prerequisite for the evaluation of GEOS-5 clouds in terms of their CVS class frequency

and the CRE statistics associated with these CVS classes is creating comparable datasets.
Assigning CVS classes to grid cell GCM cloud fields is not possible without manipulation of the
GCM's cloud profiles. To this end, we use the cloud subcolumn generator that is paired with the
RRTMG-LW and RRTMG-SW radiative transfer codes (Mlawer et al., 1997; Iacono et al., 2008) in
the model's Monte Carlo Independent Column Approximation (McICA; Pincus et al., 2003)
implementation. This subcolumn generator follows Räisänen et al. (2004) and can produce
subcolumns that are consistent with specific assumptions about the vertical overlap of both
cloud fraction and the horizontal distributions of cloud condensate. While the latter type of
overlap is irrelevant to CVS class frequency statistics, it does matter for the radiative transfer
calculations producing the radiative fluxes used to estimate CREs. The 140 subcolumns created
by the model's generator (which match the number of "g points" in RRTMG-LW's correlated-k
scheme) are essentially assumed equivalent to the cloud profiles viewed by the active
instruments (CALIPSO's lidar and CloudSat's radar) and whose vertical location information is
recorded in the 2B-CLDLASS-LIDAR product. Herein, we will show results from two types of
cloud fraction overlap schemes that have been implemented in the cloud subcolumn generator,
generalized (GN) overlap, also known as exponential overlap (Hogan and Illingworth, 2000;
Oreopoulos and Norris, 2011) and maximum-random overlap (MR overlap, Geleyn and
Hollingsworth, 1979).
The model, GEOS-5 tag "Jason-2_0" was run with fixed sea surface temperatures (SSTs)
for the same period as the reference dataset, 2007-2010. The model integration was driven by
radiative fluxes and heating rates produced by applying generalized overlap in the radiation
calculation. RRTMG-LW and RRTMG-SW were called for an additional set of flux calculations
using this time the MR overlap assumption to produce cloudy subcolumns, but only in
diagnostic mode, i.e., the generated fluxes served only diagnostic purposes and were not
passed back to the model to influence the evolution of its energetics and dynamics. This way,
with one interactive and one diagnostic call to the RRTMG codes, we were able to obtain two
sets of CVS diagnostics and corresponding CREs. In both cases, the subcolumns come from a
common mean cloud fraction and condensate profile. The layer condensates are assumed to
possess horizontal subgrid condensate heterogeneity as prescribed in Oreopoulos et al. (2012).
This subgrid condensate variability affects the model's CRE distribution, but not the CVS fields
and statistics.
In the subcolumn generator, the decorrelation length (e-folding distance) for the
generalized overlap scheme was set to vary zonally as described in Oreopoulos et al. (2012).
The physical meaning of the decorrelation length is that cloud layers separated by a distance
equal to the decorrelation length overlap as a mixture of maximum and random overlap in $e$
(≈0.368) and 1-$e$ (≈0.632) proportions (weights), respectively. At distances greater (smaller)
than the decorrelation length the contribution of random (maximum) overlap contribution
increases (decreases) compared to the above values. In the limit of zero separation cloud
overlap is purely maximum, while in the limit of infinite distance overlap is purely random. The
zonal prescription of decorrelation length by Oreopoulos et al. (2012) is based on CloudSat
observations and is meant to capture a more coherent vertical cloud alignment (i.e., more
maximum overlap and greater decorrelation length) at low latitudes compared to high
latitudes, also seen by Barker (2008). This formulation of overlap is an alternative to maximum-
random overlap which was the standard popular choice in earlier years. The Geleyn and
Hollingsworth (1979) implementation of MR overlap in our Räisänen et al. (2004)-based
generator allows for random overlap even within a "block" of contiguous clouds: immediately
adjacent clouds are maximally overlapped, but non-adjacent clouds within the contiguous block
can have portions that are randomly overlapped if there is a local minimum in cloud fraction
between them; random overlap applies for those cloudy portions that do not fully overlap with
the in-between layers. This type of MR overlap should be contrasted with other
implementations (e.g., Chou et al., 1998) where maximum overlap always takes place within
the block while the various distinct blocks of the atmospheric column (always separated by
clear layers) are themselves randomly overlapped.
**3. GEOS-5 cloud evaluation with CVS**
*a. Climatological CVS occurrence*
Figure 2 compares the observed and simulated (from GN overlap) multi-annual maps of
the relative frequency of occurrence (RFO) for all seven CVS classes of our study. The observed
fields are sampled at rather coarse 4°x4° scales to compensate for the substantial sparseness of
the active observations (gridding at higher resolutions would make for relatively noisy maps).
Above each panel, we provide the area-weighted RFO global mean of the CVS (equivalent to its
global cloud fraction). These fields include nighttime observations and simulations since the
former are possible for active sensors and the latter are passed as input for the model's
nighttime RRTMG-LW calculations.
Before examining consistency (or lack thereof) for cloud fields, we first turn our attention
to clear skies. We note that the observations suggest a cloudier world with clear skies occurring
only ~25% of the time (or, alternatively, covering 25% of the global area between 82°S and
82°N). The GEOS-5 AGCM on the other hand produces clear skies more frequently, ~38% of the
time over the entire globe (90°S to 90°N) for GN and ~42% for MR. Despite the model's positive
clear-sky fraction bias (negative cloud fraction bias), many patterns of clear-sky occurrence are
realistic with peaks occurring in desert areas, western North America and the southern parts of
Africa and S. America. Over the ocean, the model seems to be producing clear skies in the
Maritime Continent and the far southern oceans more frequently than observations, but these
overestimates are still much smaller compared to those in wide subtropical swaths of the
Atlantic and Pacific oceans. The model also exhibits pronounced cloudiness underestimates in
the descending branch of the central Pacific's Walker circulation. The only notable model
underestimate of clear-sky frequency occurs over western Antarctica. The MR overlap
assumption makes the clear sky overestimates worse, with the biggest impact seen in the
central and western tropical Pacific (clear subcolumn panel of Fig. 3). Note that the observed
global clear sky fraction is lower in 2B-CLDCLASS-LIDAR compared to passive satellite
observations such as those from MODIS (King et al., 2003) because of CALIOP's enhanced ability
to detect clouds that are optically very thin. Model cloud coverage on the other hand has been
traditionally tuned to resemble that seen in cloud climatologies obtained by satellite
observations from passive imagers at solar and thermal infrared wavelengths.
Moving on to cloudy skies, a quick survey of the remaining panels of Fig. 2 reveals that the
model exhibits considerable skill in simulating cloudiness when viewed under the prism of CVS
classes. Weaknesses become however apparent upon closer examination. In terms of global
values, the only CVS class where the model produces a substantial RFO overestimate is "HM",
for both overlap assumptions. For CVS = "HML", global RFOs agree, especially for the GN
overlap assumption. The global RFOs of all other CVS classes are underestimated to varying
degrees with the underestimates being slightly worse for the MR overlap assumption, except
for CVS = "L" for which MR RFO slightly exceeds GN RFO. The total RFO of the four CVS classes
containing $H$ clouds is ~40% in observations, and ~36% (GN) or ~32% (MR) in the model. The
remaining CVS classes consisting of only $L$ and $M$ clouds add up to a global RFO of ~35% in
observations and ~ 26% in the model (both GN and MR). Therefore, most of the 13%
discrepancy between GEOS-5/GN in global cloud fraction comes from the three CVS classes
containing only $L$ and $M$ clouds, while the larger discrepancy of ~17% for GEOS-5/MR is more
evenly split between these three CVS classes and the remaining four containing $H$ clouds.
A closer comparison of geographical features is also informative. The bottom part of Fig. 2
shows only the GN overlap results and can be directly compared with the top part showing the
observed maps. The performance of the MR overlap implementation can be gleaned in terms of
its deviation from GN in the Fig. 3 difference maps.

Simulating low clouds has been identified as a challenge for large-scale models, but this version of GEOS-5 seems to be simulating the isolated low clouds (CVS = "L") quite well with a global underestimate of ~5% for GN overlap and ~4%, for MR (absolute values), and with characteristic cloud patterns associated with marine stratocumulus being present albeit with less extensive spatial coverage. While GEOS-5 does not produce isolated *M* clouds (CVS class "M") as often as in the observations, the impact is expected to be small as this CVS class is the least frequently observed and with presence exclusively over land and specifically deserts, ice/snow covered surfaces, and regions of pronounced orography. Overall however, there is not such a great paucity of *M* clouds in the model when taking into account the other CVS classes containing this type of cloud. Setting aside deep and multi-layer clouds (the "HML" CVS class), *M* clouds appear only about 11% (for GN – the figure rises to 22% for MR) more frequently (in relative terms) in observations than the model; the combined RFO of "M", "ML" and "HM" is 14.5% in the observations and 13% (11.8%) in the model for the GN (MR) implementation. Finally, *H* over *L* clouds (CVS class "HL") are one of the biggest contributors in the overall cloudiness discrepancy between the real and simulated worlds as they appear twice as often in the observations than in the GN version of the model (and even more relative to the MR implementation of the model). The model seems to be lacking much of the tropical presence of this CVS class.

A closer look at the influence of the overlap assumption on CVS RFOs can be gauged from the Fig. 3 maps. We have previously seen that the MR overlap assumption generally produces less cloudiness than GN. This happens systematically (i.e., virtually all locations) for five out of seven CVS classes. The interesting exception is CVS = "L" (CVS = "M" is absent in GEOS-5 for all practical purposes). The Fig. 3 difference map for CVS = "L" reveals that the GN's reduced cloudiness comes mostly from the extratropics; tropical and subtropical pockets can be found where GN cloud amounts exceed those from MR, as in the other CVS classes. The contrast between CVS = "L" and the other CVS classes illustrates the fact that the specific flavors of these overlap assumptions as implemented in GEOS-5 can produce a variety of outcomes that depend on the total geometrical extent of contiguous or non-contiguous cloud vertical cloud configurations and the detailed shape of the cloud fraction profile.

*b. Global CRE comparison by CVS class*

Figure 4 compares the global mean CRE between model and observations, the latter
($\bar{r}$), coming from the aforementioned 2B-FLXHR-LIDAR CC product. It shows the mean values
only *when the CVS occurs*, i.e., CRE is weighted by area, but not by global RFO. We call this type
of CRE the "cloudy-column" or "overcast" CRE since it is calculated by taking the mean of the
CRE values of cloudy subcolumns belonging to the CVS class. CRE values for each cloudy
subcolumn also correspond to overcast conditions since there is no partial cloudiness at the
subcolumn scale. We show overcast CRE from three perspectives: the Top-of-the-Atmosphere,
TOA (Figure 4a), the surface, SFC (Figure 4c), and the atmospheric column, ATM (Figure 4b), the
latter derived as the difference between the TOA and SFC CREs. Moreover, we distinguish
between shortwave (SW) and longwave (LW) components, and also display their sum which we
call "total" CRE (aka "net" CRE). With CRE being defined as the difference between cloudy and
clear-sky net (=down-up) fluxes, negative values indicate a radiative cooling effect, while
positive values indicate a radiative warming effect. For TOA and SFC, all SW CREs are negative.
Note also the magnitudes at TOA and SFC being rather similar for SW, with the slightly larger
SFC value resulting from the small positive ATM SW CRE which indicates that clouds slightly
enhance atmospheric column absorption. While LW CREs at both TOA and SFC are positive and
therefore indicative of warming, the ATM LW CRE can be either positive or negative. Note that
all positive global means involve *H* clouds. Again, we show model results for the two overlap
assumptions, GN and MR although their CREs are quite close in general. The observed SW CREs
depend strongly on the incoming solar flux at the approximate 1:30 pm local overpass time, and
are therefore scaled to diurnal fluxes by normalizing with the ratio of the instantaneous to
diurnally averaged incoming solar flux at TOA (O17); the LW CREs are simple averages of the
daytime and nighttime overpass values. On the other hand, both SW and LW CREs from the
model are daily averages of three-hourly mean outputs.
For TOA SW CRE, the best agreement between model and observations occurs for CVS =
"L", and CVS = "HM". For the remaining CVS classes the model either overestimates (CVS = "H",
"M", "HL") or underestimates (CVS = "ML", "HML") overcast TOA SW CRE. The overestimate for
CVS class "H" is very large in relative terms given the small absolute magnitude of the observed
CRE. It appears then that *H* clouds in the model are optically thicker than in observations.
Discrepancies are quite smaller for TOA LW CRE, reflecting the lesser dependence of this
quantity on cloud properties other than cloud top location (which is constrained because of the
CVS class decomposition) once clouds reach a certain value of optical thickness (~ 5). The
biggest bias (underestimate) appears for CVS = "HML" CVS, but since it is still smaller than the
SW CRE bias it results in an underestimate of net planetary cooling as expressed by total TOA
CRE (purple bars). Given the better agreement between LW CREs, total TOA CRE biases largely
follow the sign of the SW CRE biases. These findings are very insensitive to the type of chosen
overlap, although the differences in magnitudes between the two simulated values are still
large enough to be distinguishable in most cases.
When moving to examination of surface (SFC) CREs (bottom panel of Fig. 4) our
conclusions about the SW CRE component are the same as before since atmospheric (ATM) SW
CREs are small positive values (middle panel). LW CRE values are again simulated quite well
since most of the variability is driven by the location of the cloud bottom which is constrained
by CVS class. The largest biases occur for CVS = "L" and "HL" (overestimates by the model), and
since the TOA CREs have small biases for those cases, errors (excessive cooling) materialize in
the ATM LW CRE. Still, the largest ATM LW CRE error occurs for CVS="HM" (excessive warming
by the model) because the TOA and SFC CRE errors are in the opposite direction. Given the
small magnitude of ATM SW CRE, the total ATM CRE errors track those of the LW component.
Figure 5 compares observed and modeled CRE values that are now weighted by the global
mean RFO ($\bar{f}$ for observations) of the CVS classes in addition to areal weighting. We can call this
type of CRE "all-sky" CRE since in the calculation of the mean all subcolumns that do not belong
to the CVS class under consideration contribute zero errors. Summing then these CVS-specific
values yields the true global CRE of observed and modeled CRE fields. Since the all-sky CRE
values and the range of the y-axis are much smaller than in Fig. 4, it makes sense to compare
the two figures only with respect to relative biases, essentially focusing on the position of the
symbols (simulated values) relative to the bar (observed values). While this will be shown more
explicitly in upcoming Fig. 6, comparison of Figs. 4 and 5 basically indicates whether RFO errors
suppress (i.e., compensate for) or amplify cloud property only errors. Take CVS = "HL" for
example: RFO errors (underestimates) help suppress the TOA and SFC SW (and total)
overestimates. In general, we do not see much of the opposite effect, i.e., an amplification of
relative error CRE when moving from overcast to all-sky CRE. Of course, a very low RFO also
makes an overcast CRE that previously seemed substantial to disappear, with CVS =" M" being a
characteristic case in point. The discussion of all-sky CRE error interpretation continues in the
next subsection where a more formal error decomposition framework is introduced.
*c. CRE error decomposition*
Figure 6 shows the decomposition of GEOS-5 all-sky CRE global errors $\Delta CRE$ of Fig. 5 to
overcast CRE and RFO error contributions for the GN case only (conclusions remain the same
for MR). The decomposition can be expressed as follows (e.g., Tan et al. 2015):
$\Delta CRE = \bar{f} \times \Delta r + \bar{r} \times \Delta f + \Delta r \times \Delta f$           (1)
This representation of CRE error arises when the model global all-sky CRE of a CVS class
(Fig. 5) is expressed as the product of a deviation $\Delta r$ from the observed mean overcast CRE
$\bar{r}$, (Fig. 4), and the model global RFO is expressed as a deviation $\Delta f$ from the observed mean
RFO, $\bar{f}$:
$CRE_{GEOS-5} = (\bar{r} + \Delta r) \times (\bar{f} + \Delta f)$           (2)
Basically, the model's grid-mean CRE error for a CVS class arises from a combination of
overcast CRE bias $\Delta r$ under the observed RFO $\bar{f}$, and the simulated RFO bias $\Delta f$ under observed
overcast CRE $\bar{r}$, plus a co-variation term of RFO and CRE errors under observed $\bar{f}$ and $\bar{r}$ (Tan et
al. 2015). Such a decomposition of CRE error allows us to infer, for example, whether the
model's poor simulation of all-sky CRE is mostly due to errors in simulating the occurrence
frequency of the CVS class or errors in the optical and physical properties of the CVS class which
drive the overcast CRE. Similarly, it potentially reveals cases where good simulations of global
all-sky CRE in Fig. 5 benefit from compensating errors in simulated RFO (Fig. 2) and overcast CRE
(Fig. 4).
Separate panels are used in Fig. 6 for SW (first column), LW (second column), and total
(third column) CRE. The breakdown by TOA, SFC and ATM is also preserved, yielding thus a total
of nine panels. In the SW, TOA (Fig. 6a) and SFC (Fig. 6g) results look again very similar, while
the ATM CRE errors (Fig. 6d) are too small to merit discussion. For most CVS classes (five out of
seven) all-sky SW CRE errors (gray bars) come from overcast CRE errors (red bars), namely
errors in CVS optical properties. The excessive planetary cooling of the cloudy columns
(negative red bars, four CVS classes) is always dampened by compensating errors, sometimes
virtually eliminating the error (as in CVS = "L", "HL") or reducing it slightly (CVS = "H"), or
overcorrecting (CVS = "M"). SW TOA overcast CRE (red bars in Fig. 6a) in the opposite direction
(cooling underestimates) become bigger all-sky errors due to RFO errors for CVS= "ML" and
"HML", while the all-sky errors for CVS = "HM" come almost exclusively from RFO errors (blue
bars in Fig. 6a). Finally, three CVS classes have sizable co-variation errors (green bars) in the
same direction as RFO errors. The above error interpretation applies virtually intact for surface
(SFC) SW errors (Fig. 6g).
Contrary to the SW, the LW CRE errors for all three vantage points (TOA in Fig. 6b, SFC in
Fig. 6h, ATM in Fig. 6e) deserve their own discussion as they have different characteristics. At
TOA and SFC, the errors are substantially smaller than their SW counterparts. Three of the four
CVS classes with $H$ clouds (the exception being CVS = "H") exhibit ~2 Wm$^{-2}$ (absolute) errors,
coming from RFO contributions in two out of the three classes. These three classes have smaller
all-sky errors at the SFC, in one case (CVS ="HL") because of compensating errors. The largest
component errors occur for CVS = "L" which has the largest absolute magnitude of all-sky SFC
CRE, but with component errors in the opposite direction, compensation reduces the all-sky
CRE error. Because the TOA errors for this CVS class are small, the SFC errors carry to the ATM
errors. The other CVS class with large ATM error is "HML" where TOA and SFC errors of the
opposite sign conspire to magnify the ATM error.
Errors in total all-sky CRE are driven mainly by SW errors at TOA and SFC (Fig. 6c and Fig.
6i), and LW errors for ATM (Fig. 6f). Errors of the opposite sign reduce the overcast cooling
error at the SFC for CVS = "L" and "HL" and the all-sky warming error for CVS="ML". But
because the SFC LW CRE errors are in general small, the total CRE SFC errors largely retain the
characteristics of the SW component. In the atmospheric column, SW and LW overcast (and all-
sky) errors are additive for CVS = "HML" and opposing for CVS = "L", the only two classes for
which ATM SW CRE registers errors of notable magnitude (cf. middle panels of Fig. 4 and 5).

4         In summary, this decomposition analysis showed the multiple ways relatively good

agreement with observed all-sky CRE values from various vantage points can be achieved by
GEOS-5 (or any other model evaluated this way). Overcast CRE and RFO errors can compensate,
TOA and SFC all-sky CRE errors can compensate (for ATM LW CRE, e.g., CVS = "M", "ML"), SW
and LW errors can compensate for total CREs, and finally the errors among various CVS classes
can compensate towards decreasing the global CRE error.
*d. Seasonal CRE comparison*
Figures 7-9 compares the multi-year mean annual cycle of TOA, SFC, and ATM total (=SW+LW)
all-sky CRE zonal averages between observations and the model (employing the GN overlap
assumption) for the four CVS classes with the greatest all-sky SW or LW CREs according to Fig.

15     5.

16         Inspection of the TOA and SFC CRE plots shows that the model has some skill in simulating

the seasonal competition between SW and LW CRE, but this should not come as a surprise as it
is driven mainly by seasonal changes in insolation. Basically, with everything else staying the
same, the SW CRE contribution to total CRE scales with the amount of incoming solar energy.
Positive values of total TOA and SFC CRE occur when the solar insolation is weak during the
winter allowing thus the positive LW CRE to exceed the negative SW CRE. At TOA, this takes
place only for the "HML" CVS class since this is the class with competing SW and LW CREs of
relatively large magnitude. Note that the model summer planetary cooling is stronger than in
the observations. At the SFC, besides CVS = "HML" the seasonal switch from cooling to warming
also takes place for CVS = "L" because the LW CRE is of comparable magnitude to its SW
counterpart. The model's CVS = "H" is virtually neutral radiatively at TOA throughout the year,
in contrast to the observations, where it provides planetary radiative heating in the tropics and
subtropics. It seems then that in the model CVS= "H" consists of optically thicker clouds that
reflect more solar radiation to space than in the real world. *H* clouds in GEOS-5 appear to be
also optically thicker when overlapping with *L* clouds (CVS = "HL"), in this case producing
planetary cooling in the tropics throughout the year and in the extratropics during the summer
months of high insolation, in contrast to the observations where their cooling effect is very
weak and occurs only in the austral extratropics during summertime. Evidence for optically
thicker *H* clouds in both CVS ="H" and "HL" is also seen at SFC total CREs which are more
negative in the model than in the observations. Overall (all CVS classes combined, the rightmost
panel of Figs. 7 and 8), the model produces a rather realistic pattern of seasonal variations in
zonal mean total CRE.
Total ATM CREs are driven as we have seen earlier by the LW component, and their
seasonal cycles are fairly well represented by the model for three of the four most radiatively
important CVS classes (Fig. 9). The nature of CVS = "HML" seems however to be different in
GEOS-5 compared to observations. At high latitudes, the atmospheric column is cooled by this
type of cloudiness, especially during the summer months, as the SFC total CRE (Fig. 8) exceeds
the TOA CRE (Fig. 7). Since the SW contribution is relatively small, it then seems that *L* clouds
within CVS = "HML" have lower bases or are optically thicker during the summer months in the
model compared to observations, making their downward emission towards the surface, and
therefore also the contrast between TOA and SFC emission, stronger in the model than the
observations. Fig. 9i shows also that the near-zero total ATM CRE for CVS ="HML" in GEOS-5
(Fig. 5) is a result of positive and negative total ATM CRE regional compensations. Overall, the
model captures the basic zonal pattern of atmospheric heating and warming (rightmost panel
of Fig. 9) with heating prevailing in the tropics and cooling in the extratropics. The tropical
heating is however weaker than in the observations while the extratropical atmospheric cooling
is stronger.
**4. Conclusions**
We have introduced a method of cloud evaluation for large-scale atmospheric models that
focuses on the vertical structure of cloudiness. Cloud Vertical Structure (CVS) is resolved in a
rather simplified way based on the various combinations of cloud presence in three standard
layers that have been traditionally used to distinguish between high, middle, and low clouds. A
reference dataset for such CVS classification now exists because of CloudSat and CALIPSO active
sensor observations (Oreopoulos et al. 2017). For the purposes of model evaluation, the initial
dataset of 10 CVS classes was simplified to consist of 7 classes by merging of some of the
original classes that had clouds in adjacent standard layers. Beyond comparison of the
frequency of occurrence of the CVS classes we also compared their radiative impact in terms of
the Cloud Radiative Effect (CRE). While the CVS classes by design constrain cloud vertical
location (albeit not in the strictest of ways), they constrain extinction to a lesser extent, and
mostly qualitatively (e.g., multi-layer cloud configurations are expected to have a greater total
column extinction). This is taken into account when examining the performance of the model in
terms of SW and LW CRE. We developed a framework wherein we can compare CRE for only
when a CVS class occurs (overcast CRE), or perform a comparison that also accounts for how
frequently the CVS class occurs (all-sky CRE). We can then naturally examine to what extent
errors in the latter type of CRE come from errors in the overcast CRE of the class and/or biases
in the frequency of occurrence.

15       The GEOS-5 model under evaluation produces about 50% more clear skies than

observations in relative terms. It produces isolated high clouds (cloud top and base above the
440 hPa level) that are slightly less frequent than in observations, but are optically thicker
yielding excessive planetary and surface cooling. Low clouds (cloud tops and bases within the
lowest layer of the troposphere up to 680hPa) are usually a challenge for global models, but
GEOS-5 is doing reasonably well and compensates a lower frequency of occurrence (by ~20% in
relative terms) with overestimates in extinction, producing in the end an excellent agreement
with observations for SW and LW all-sky CREs at either the TOA, SFC or the atmospheric column
vantage points. Overall LW CREs are better simulated since they are mainly driven by vertical
cloud location which is substantially constrained when clouds are broken by CVS class. But
either component of CRE can be off in terms of contribution to the global CRE if the frequency
of occurrence is deficient. The other side of the coin is, of course, that incorrect simulation of
the frequency of occurrence can compensate for biased cloud optical and physical properties
that determine the overcast CRE of the CVS class. Needless to say, CRE biases among different
CVS classes can also cancel out to various degrees when global or regional CREs encompassing

all clouds represented by the CVS classes are calculated. In such a holistic view, the model appears able, for example, to reproduce the aggregate planetary feature of atmospheric radiative warming in the tropics and cooling in the extratropics driven by cloud configurations dominated by high and low clouds, respectively, albeit with magnitudes that differ from those observed.

The evaluation we conducted requires that the model has the capability to produce cloudy subcolumns which are then considered equivalent to the atmospheric column profiles seen by the active observations. There is no unique way to go from mean cloud fraction profiles to subcolumns having layer cloud fractions that are either one or zero. We tried two ways to produce subcolumns that assume different cloud fraction overlaps and obtained rather close results. By adopting our framework of cloud evaluation, which, incidentally, should be used in conjunction with other cloud evaluation methodologies (e.g., cloud regimes as in Jin et al. 2017a, b), one can assess whether other large scale models are more sensitive (i.e., produce a greater diversity of CVS climatologies) to different overlap assumptions applied to the same original mean cloud fraction profiles. What one should always keep in mind however is that no matter how good the cloud subcolumn generator is, observed CVS class global frequencies and patterns cannot be reproduced if the model's underlying mean cloud profiles used as input to the generator are deficient.

**Code availability**

The GEOS-5 source code is available under the NASA Open-Source Agreement at

http://opensource.gsfc.nasa.gov/projects/GEOS-5/.

**Author contribution**: D. Lee and L. Oreopoulos designed the metrics and experiments. D. Lee adapted the model code for the new metric and performed the simulations. N. Cho processed the observational dataset. D. Lee and N. Cho created the graphics and figures. D. Lee and L. Oreopoulos authored the text with contributions from N. Cho.

**Competing interests**: The authors declare that they have no conflict of interest.

**Acknowledgments:** D. Lee gratefully acknowledges funding support from NASA's NIP program,
while L. Oreopoulos acknowledges support from NASA's CloudSat and CALIPSO Science Team
Program. Resources supporting this work were provided by the NASA High-End Computing
(HEC) Program through the NASA Center for Climate Simulation (NCCS) at Goddard Space Flight
Center. The reference data used for model evaluation (2B-CLDCLASS-LIDAR and 2B-FLXHR-
LIDAR) are available from the CloudSat Data Processing Center at
[http://www.cloudsat.cira.colostate.edu](http://www.cloudsat.cira.colostate.edu). We thank our colleague Donifan Barahona for helpful
discussions about various model tags.

**List of Figures**

**Figure 1.** The original ten CVS classes of Oreopoulos et al. (2017) used as reference for the comparison of this paper. The multi-layer CVS classes other than "HL" are merged in this paper thus reducing the total number of CVS classes to seven. We essentially do no distinguish between contiguous and non-contiguous clouds in adjacent standard layers. Dotted lines show which pairs of CVS classes have been combined for this study.

**Figure 2.** Geographical RFO distribution (%) for cloudless skies and the seven CVS classes according to CloudSat/CALIPSO observations (top 8 panels), and for GEOS-5 (GN overlap assumption, bottom 8 panels). Global mean values are shown above each panel, in the case of GEOS-5 we provide the global values for both the GN and MR overlap (in parentheses).

**Figure 3.** RFO difference (%) maps for clear skies (divided by two to use a common color scale) and the seven CVS classes as simulated by GEOS-5 using the GN and MR overlap assumptions in the cloudy subcolumn generator.

**Figure 4.** Comparison between observations and model (GN and MR) of global overcast CREs (Wm$^{-2}$): Top-of-the-Atmosphere, TOA (top), surface, SFC (bottom), and atmospheric column, ATM (middle) derived as the difference between the TOA and SFC CREs. CREs are distinguished into shortwave (SW) and longwave (LW) components, and their sum, "total" CREs for each CVS class are also shown. Note that the y-axis range is the same for TOA and SFC CRE, but it is substantially more compressed for ATM CRE.

**Figure 5.** As Fig. 4, but for all-sky (RFO-weighted) CREs.

**Figure 6.** Decomposition of all-sky CRE error (Eq. 1) for GEOS-5 CVS classes when the GN overlap assumption is used. Gray bars represent the overall all-sky CRE error, and the remaining bars contributions to that error as follows: red bars represent overcast CRE errors, blue bars RFO errors, and green bars co-variation errors. The nine panels represent all combinations of CRE, namely SW, LW, total at TOA, SFC and within ATM.

**Figure 7.** Comparison of the multi-year annual cycle of TOA, total (=SW+LW) all-sky CRE zonal averages (W/m$^2$) between observations (top row, panels a to e) and the model (bottom row, panels f to j) when employing the GN overlap assumption for the four CVS classes with the

greatest all-sky CREs according to Fig. 5. The rightmost set of panels displays the scaled (half)
total CRE of all CVS classes combined.
**Figure 8.** As Fig.7, but for SFC total all-sky CRE.
**Figure 9.** As Fig. 7, but for ATM total all-sky CRE.

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

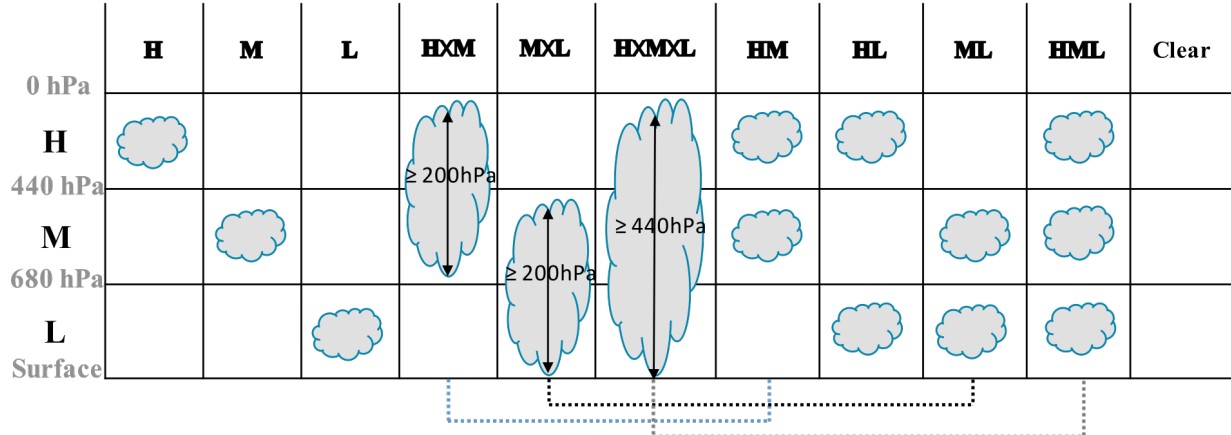

**Figure 1**. The original ten CVS classes of Oreopoulos et al. (2017) used as reference for the
comparison of this paper. The multi-layer CVS classes other than "HL" are merged in this paper
thus reducing the total number of CVS classes to seven. We essentially do no distinguish
between contiguous and non-contiguous clouds in adjacent standard layers. Dotted lines show
which pairs of CVS classes have been combined for this study.

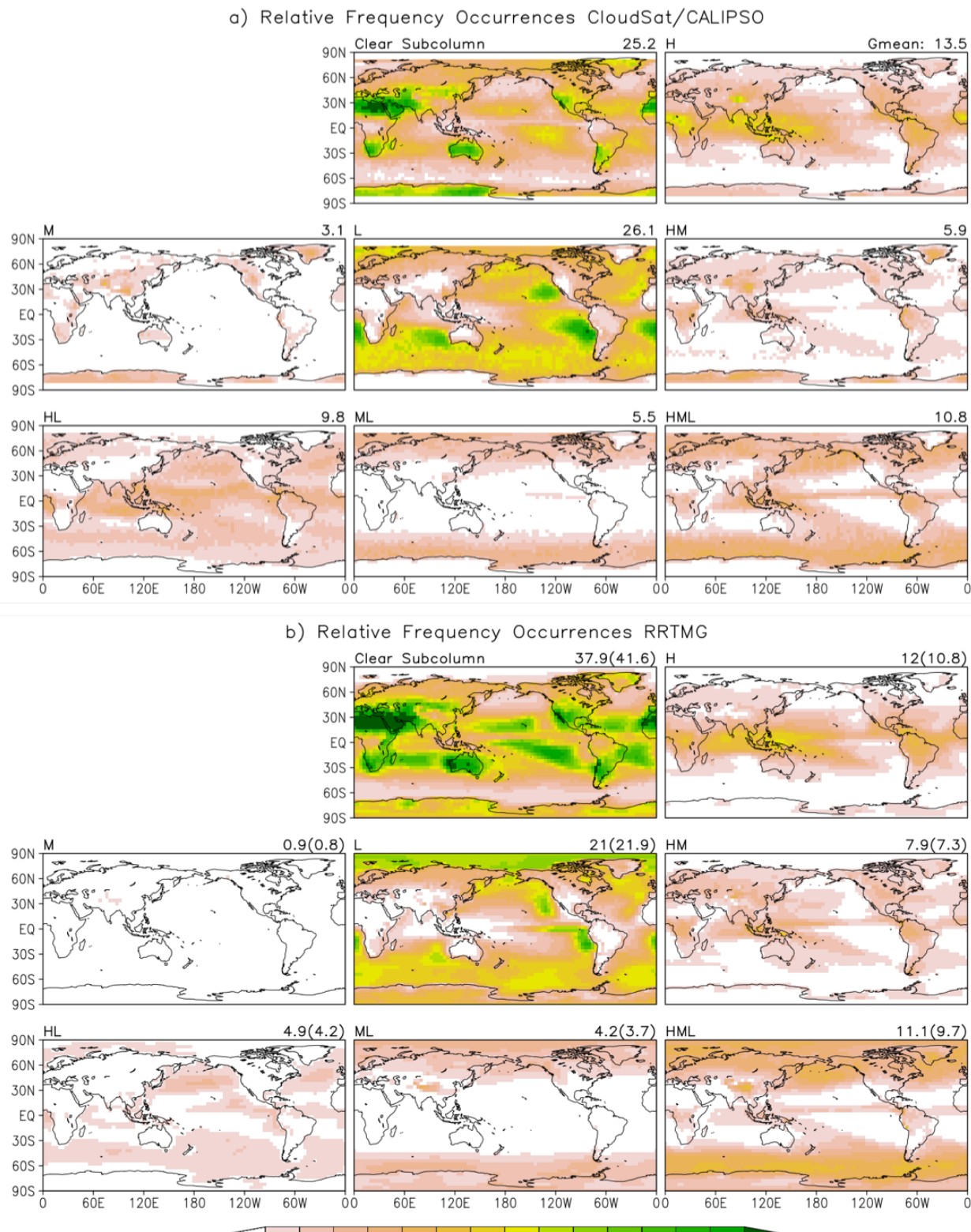

2   **Figure 2.** Geographical RFO distribution (%) for cloudless skies and the seven CVS classes

3   according to CloudSat/CALIPSO observations (top 8 panels), and for GEOS-5 (GN overlap

assumption, bottom 8 panels). Global mean values are shown above each panel, in the case of
GEOS-5 we provide the global values for both the GN and MR overlap (in parentheses).

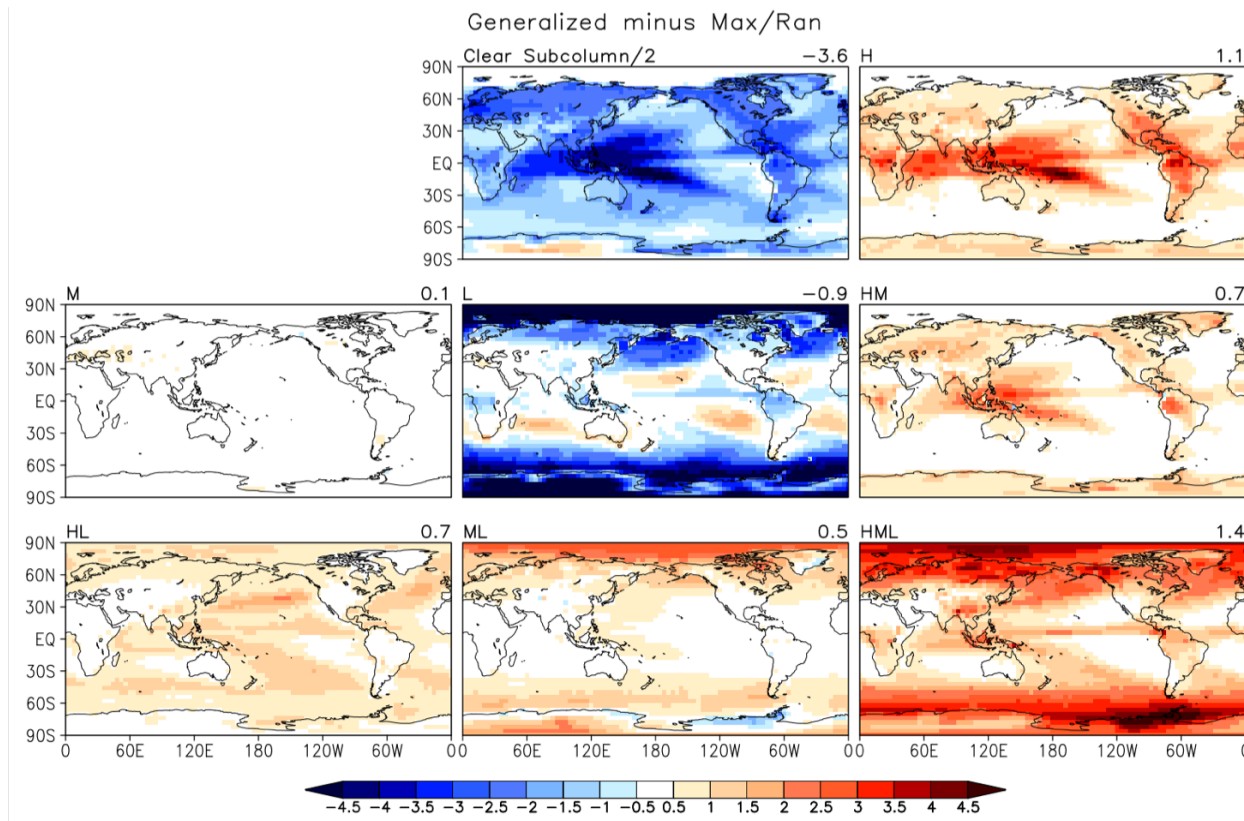

**Figure 3.** RFO difference (%) maps for clear skies (divided by two to use a common color scale)
and the seven CVS classes as simulated by GEOS-5 using the GN and MR overlap assumptions in
the cloudy subcolumn generator.

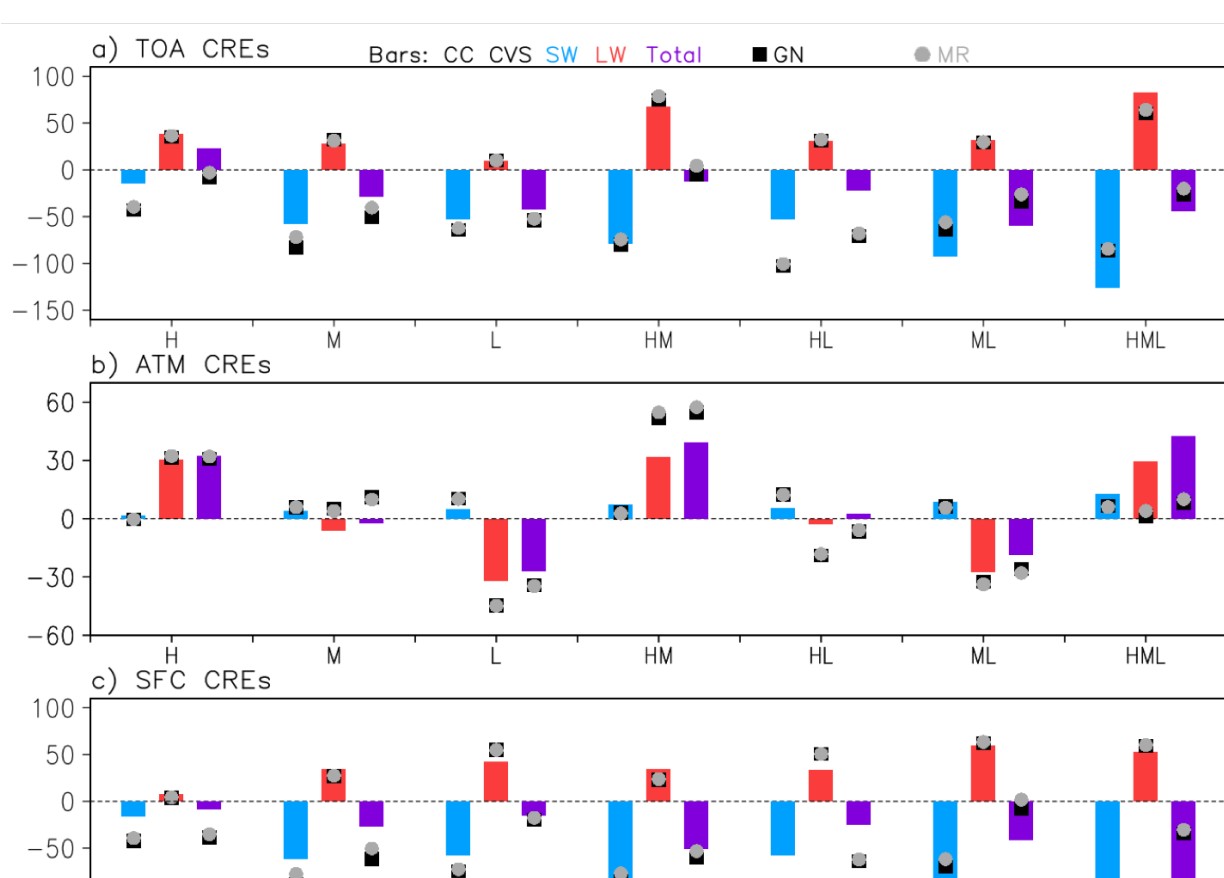

**Figure 4.** Comparison between observations and model (GN and MR) of global overcast CREs

(Wm$^{-2}$): Top-of-the-Atmosphere, TOA (top), surface, SFC (bottom), and atmospheric column,

ATM (middle) derived as the difference between the TOA and SFC CREs. CREs are distinguished

into shortwave (SW) and longwave (LW) components, and their sum, "total" CREs for each CVS

class are also shown. Note that the y-axis range is the same for TOA and SFC CRE, but it is

substantially more compressed for ATM CRE.

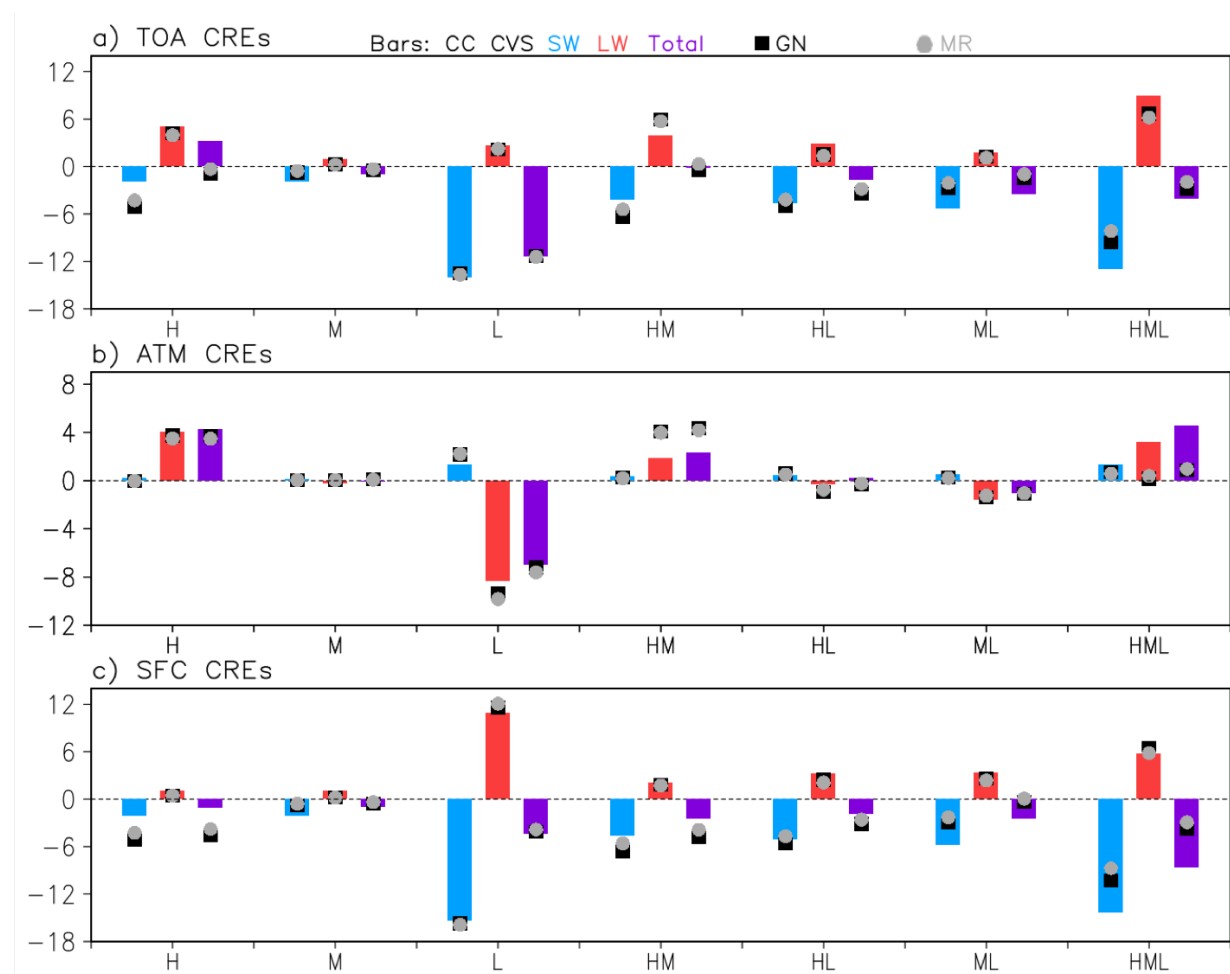

3 **Figure 5.** As Fig. 4, but for all-sky (RFO-weighted) CREs.

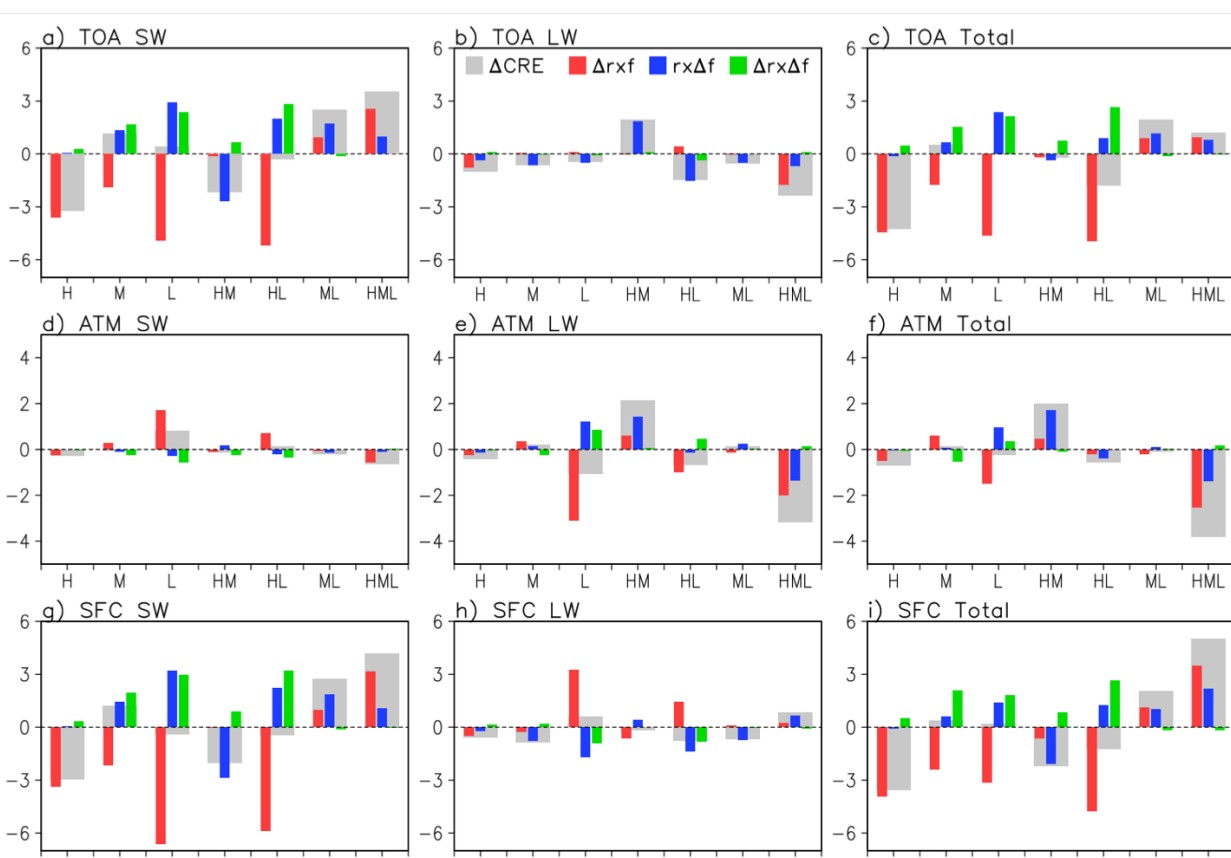

**Figure 6.** Decomposition of all-sky CRE error (Eq. 1) for GEOS-5 CVS classes when the GN

overlap assumption is used. Gray bars represent the overall all-sky CRE error, and the remaining

bars contributions to that error as follows: red bars represent overcast CRE errors, blue bars

RFO errors, and green bars co-variation errors. The nine panels represent all combinations of

CRE, namely SW, LW, total at TOA, SFC and within ATM.

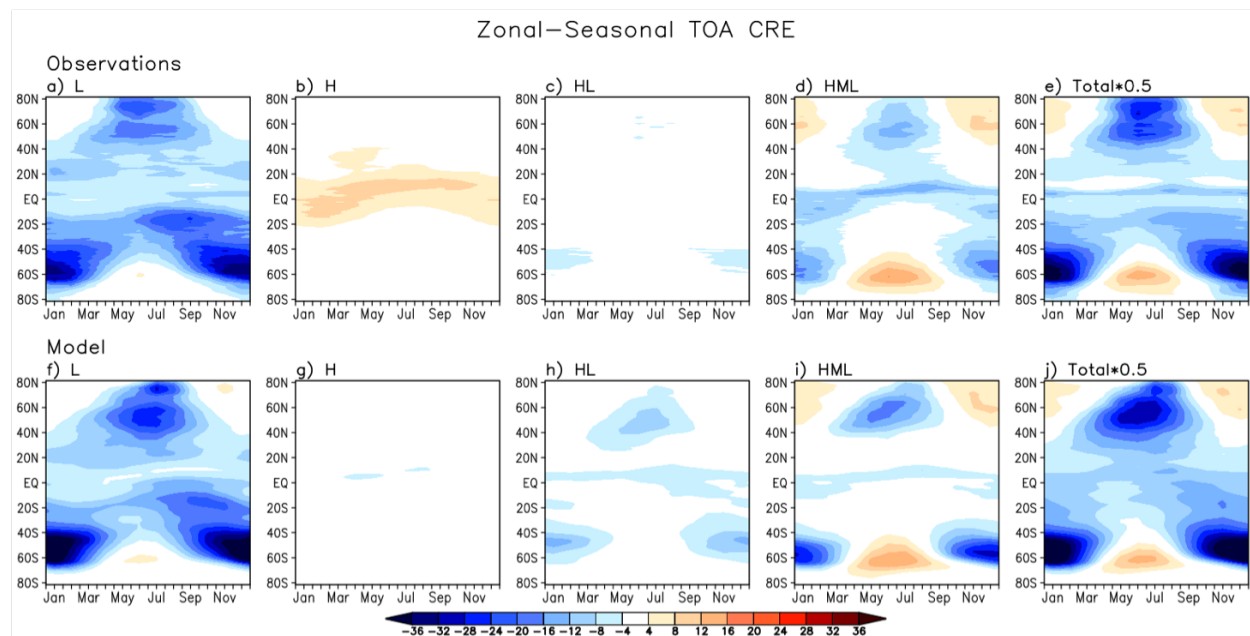

**Figure 7.** Comparison of the multi-year annual cycle of TOA, total (=SW+LW) all-sky CRE zonal
averages (W/m$^2$) between observations (top row, panels a to e) and the model (bottom row,
panels f to j) when employing the GN overlap assumption for the four CVS classes with the
greatest all-sky CREs according to Fig. 5. The rightmost set of panels displays the scaled (half)
total CRE of all CVS classes combined.

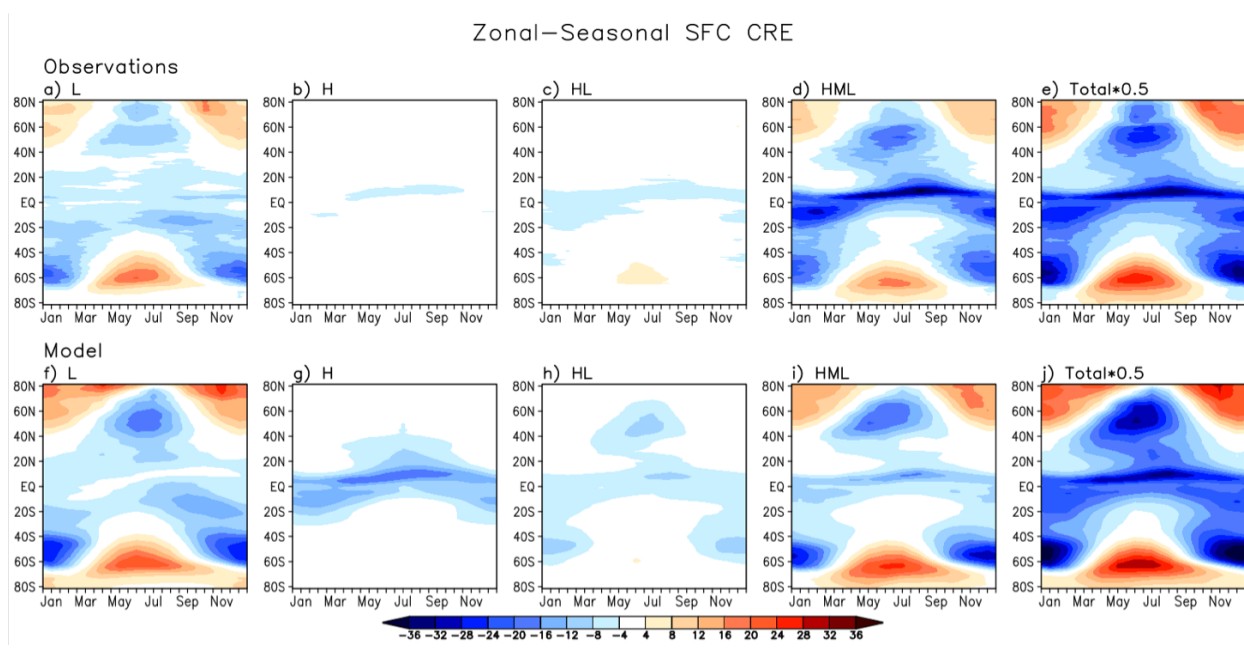

2    **Figure 8.** As Fig.7, but for SFC total all-sky CRE.

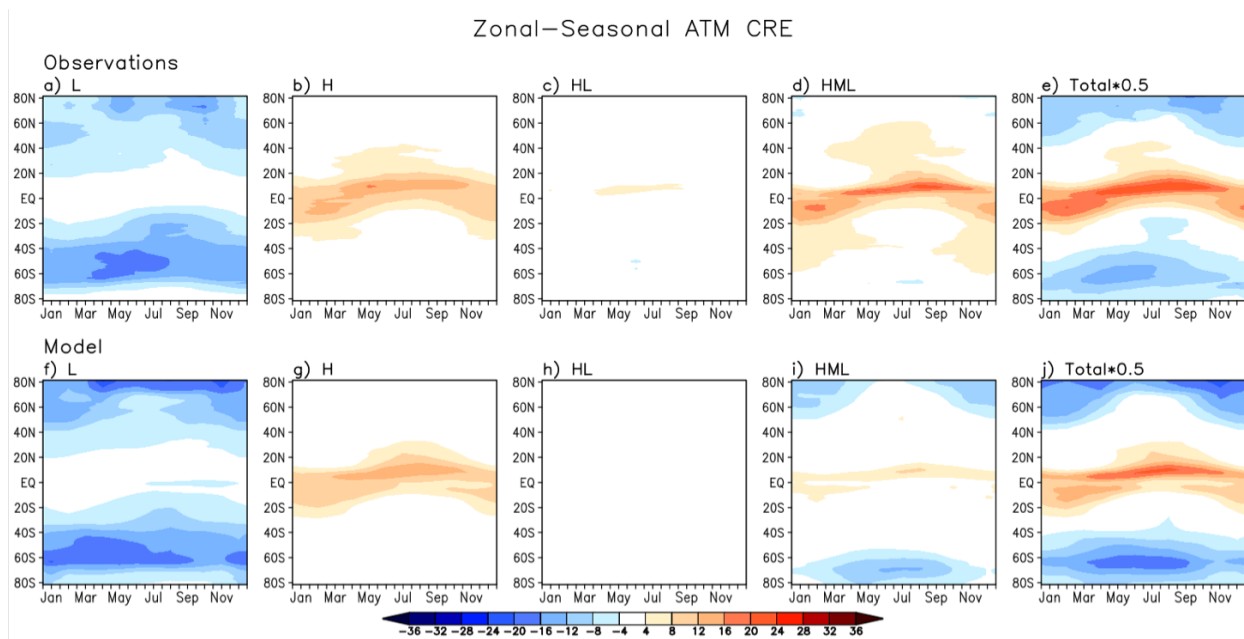

Figure 9. As Fig. 7, but for ATM total all-sky CRE.