# Peer review of "An evaluation of clouds and radiation in a Large-Scale Atmospheric Model using a Cloud Vertical Structure classification"

_Geoscientific Model Development, 2019_

## Referee Comment (RC1) · Anonymous Referee #1 · 30 Oct 2019

Disclaimer: This is my first review of a GMD manuscript. In addition, my background is in atmospheric observations rather than numerical modelling. These circumstances might be reflected in my comments.

Lee at al. describe a method to assess the representation of clouds and their radiative effects in a global model based on A-Train observations and sub-column generators. At least the latter is not within my field of expertise and I had quite some difficulties in understanding how these sub-column generators work. However, I'd leave the assessment of the model-related methodology to an expert reviewer. In terms of presentation, there are a couple of items below that might make it easier for the reader to follow the

[Figure]

reasoning of the authors. Finally, the use of jargon and abbreviations quite affect the readability of the text.

Detailed comments:

It has not been clear to me from reading the abstract what this paper is about, particularly in terms of quantitative findings.

page 4, line 12/13: Cloudsat and CALIPSO have collected a much longer time series. Wouldn't it be better to base the analysis on 10+ years of observations?

page 5, lines 4-6: To me the fact that the model produces more than the generally observed four cloud layers sounds like there might be an issue with properly representing clouds in the model. Please comment.

page 5, line 10 - page 7. line 3: I'll leave commenting on this part to modelling experts. Suffice to say that I don't understand from this text how the two sub-column generators work.

page 7, line 21: not sure maritime continent is used correctly here, but I might have mixed up plots

page 9, line 16: If I understand correctly, this is a comparison of the findings of the two sub-column generators. What is your benchmark for stating that one method produces underestimates?

page 10, lines 1-4: This statement should be moved to the figure caption. Also, state at the start of the discussion what kind of averaging has been applied.

page 11, discussion of Figure 5: Do I understand correctly that Figure 4 refers to values for all instances at which the respective cloud types are present while Figure 5 presents those numbers normalised by the occurrence rate of the respective cloud types? This information is somewhat hard to extract from the text (but might be due to use of jargon). Also, what is meant with grid-mean? I have really no idea what global

grid-mean is supposed to be.

page 12: I understand that f and r are expressions for parameters that have been used earlier in the manuscript. Why not introduce them at the first instance they are being used? It would be much less confusing if you would be consistent with the naming parameters throughout the text!

Please make use of the sub-figure labelling (a, b, c, etc.) in the discussion of the findings. This would make it much easier for the reader to follow your argument.

page 13, lines 23-28: This is a somewhat disappointing conclusion. What's the contribution of this study apart from anything's possible?

page 20, line 19: please update reference.

Figure 1: I took notice of the dotted lines in the figure only after trying to make sense of the combination of classes from your description myself. The purpose of those lines is not stated anywhere. Please add: dotted lines show which cloud classes have been combined for this study.

Figures 2 and 3: I would suggest to revise these figures in a more intuitive way. For instance, you could have columns with high clouds on top and low clouds at the bottom. Also have all cases in Figure 2a in one column, then the plots of Figure 2b next to it in another column, and then the findings of Figure 3 in a third column. However, Figure 3 seems pointless to me as it presents the differences between the two sub-column generators. It would be better to present the difference to the observations for each sub-column generator, i.e. as an individual column added to Figure 2. I would also prefer to see plots from -180 to +180 degree. Finally, please elaborate on the global mean values. Shouldn't they add up to 100%. They don't right now.

Figures 4 to 6: Please add a line at zero. See also my comments regarding the discussion on page 11. In Figure 6, you should at a statement to the caption regarding the meaning of the grey bars.

[Figure]

Figures 7 to 9: I'd suggest to refer to the two lines as observations and model.

---

## Referee Comment (RC2) · Anonymous Referee #2 · 4 Nov 2019

The simulated cloud radiative effects (CRE) are commonly biased in most climate models. This study, with the aid of both CloudSat/CAPLIPSO retrievals and the implemented subcolumn cloud generator in NASA's GEOS-5, explores the CRE biases in association with cloud vertical structure classification. Results show that while the simulation of global CRE is in much agreement with observation, the CRE due to different cloud classification is not. Moreover, by decomposing model's CRE errors into components stemming from biases in RFO and cloudy-column CRE, the relatively good simulations of global grid-mean CRE largely benefit from compensating errors in these two terms. The method introduced in this paper can be used in other models to explore their CRE behavior, thus beneficial for the modeling community.

[Figure]

While I generally find the manuscript suitable for publication in GMD, further improvements are needed before the manuscript is accepted. Below I have included a list of the major comments that I think should be addressed, followed by a list of more specific comments.

Major comments:

1. The study uses 2B-FLXHR-LIDAR
product as a guiding reference for CRE, which was obtained by invoking radiative transfer algorithm operating on thermodynamical fields from re-analysis and cloud properties from CloudSat/CAPLIPSO retrievals. As the authors mentioned, the SW CREs in 2B-FLXHR-LIDAR is strongly time dependent. It is worth to add CERES data as a reference as well, since a great many models are commonly tuned to resemble CERES observations.

2. Besides cloud overlap assumptions, the cloudiness vertical profile per se is important for the determination of CVS classification. The authors are suggested to replenish the role of layer cloud fraction when revising the paper. The CVS classification may suffer from poor representation of subgrid cloud condensation and/or overlap assumption. In addition, the vertical resolution in GCMs is typically coarser than that in CloudSat/CALIPSO retrievals, which to some extent plays an important role in calculating RFO. The authors need point out this in the paper.

3. When comparing the two overlap assumptions, the GN assumption yields more clouds than MR in almost all cloud regimes except for isolated low clouds in extratropics. Why does this occur? Is this because clouds in this region are nonadjacent separated by clear skies that have a vertical distance smaller than specified decorrelation length in GN?

Specific comments:

1. The abbreviation "RFO" is not fully spelled at the first place in the main text.

2. P9L1 regions pronounced orography -> regions with pronounced orography
 3.

[Figure]

P9L17 exceeds -> exceed 4. P27 As Fig. 3-> As Fig. 4.

---

## Author Response (AR1)

Authors' response on "An evaluation of clouds and radiation in a Large-Scale Atmospheric Model using a Cloud Vertical Structure classification" by Dongmin Lee et al.

We thank Referees and Editor for very insightful and constructive comments on our discussion paper. All the points are well taken, and we have made a number of revisions to the paper to clarify the points raised in the review. The point by point responses and description of changes are included in the authors' response. And a marked-up manuscript version showing the changes made is attached.

*Anonymous Referee #1*

*Disclaimer: This is my first review of a GMD manuscript. In addition, my background is in atmospheric observations rather than numerical modelling. These circumstances might be reflected in my comments.*
*Lee at al. describe a method to assess the representation of clouds and their radiative effects in a global model based on A-Train observations and sub-column generators. At least the latter is not within my field of expertise and I had quite some difficulties in understanding how these sub-column generators work. However, I'd leave the assessment of the model-related methodology to an expert reviewer. In terms of presentation, there are a couple of items below that might make it easier for the reader to follow the reasoning of the authors. Finally, the use of jargon and abbreviations quite affect the readability of the text.*

*Detailed comments:*
*It has not been clear to me from reading the abstract what this paper is about, particularly in terms of quantitative findings.*

A: We have revised the abstract extensively and have now added some specific results. We hope that the revised version makes now clearer what the paper is about.

*page 4, line 12/13: Cloudsat and CALIPSO have collected a much longer time series. Wouldn't it be better to base the analysis on 10+ years of observations?*

A: We use the same four-year period (2007-2010) of the merged CloudSat/CALIPSO product 2B-CLDCLASS-LIDAR R04 as our previous paper, Oreopoulos et al. (2017), for consistency. This is considered the "golden period" of the merged product as it includes both daytime and nighttime CloudSat observations. CloudSat suffered a battery anomaly in early 2011, and has been conducting only daytime observations since operations resumed.

*page 5, lines 4-6: To me the fact that the model produces more than the generally observed four cloud layers sounds like there might be an issue with properly representing clouds in the model. Please comment.*

A: First of all, complicated cloud vertical structures are being simplified in both the observation-based product and in the model. Also, the cloud scheme of GEOS-5 is not fundamentally different than counterpart schemes in similar models. The version of the model used in this study has 72 vertical layers, and clouds can form in about 30 of them. But unlike observations, stratiform modeled clouds form in the vertical layers of a column independently of each other, so that the modeled cloud profiles are typically noisier compared to the smoother vertically coherent profiles of observations. The model is basically forced to produce clouds under the limitation of an atmospheric column that is vertically discretized because the model is built on the 3D grid.

*page 5, line 10 - page 7. line 3: I'll leave commenting on this part to modelling experts. Suffice to say that I don't understand from this text how the two sub-column generators work.*

A: Subcolumn generators are standard tools in the modeling world to create subgrid cloud variability that the model does not prognosticate or otherwise resolve. They are used for both model parameterizations (radiation, cloud microphysics) and diagnostics such as cloud simulators (the COSP satellite simulator, for example, has a subcolumn generator). Subcolumn generators need rules on how to account for vertical cloud fraction overlap and cloud condensate horizontal inhomogeneity (and how it overlaps in height).

*page 7, line 21: not sure maritime continent is used correctly here, but I might have mixed up plots*

A: We refer to the oceanic regions around the Maritime Continent. In that region, clear columns are not frequently observed in CloudSat-CALIPSO, but the model seems to be producing them quite frequently.

*page 9, line 16: If I understand correctly, this is a comparison of the findings of the two sub-column generators. What is your benchmark for stating that one method produces underestimates?*

A: The reviewer's point is well-taken. This refers to just a difference between two subcolumn generators and we have therefore rephrased.

*page 10, lines 1-4: This statement should be moved to the figure caption. Also, state at the start of the discussion what kind of averaging has been applied.*

A: We have indeed expanded the figure caption using some of this text. The averaging method for the cloudy column has actually been provided before the discussion, see p. 9, lines 25-26.

*page 11, discussion of Figure 5: Do I understand correctly that Figure 4 refers to values for all instances at which the respective cloud types are present while Figure 5 presents those numbers normalized by the occurrence rate of the respective cloud types? This information is somewhat hard to extract from the text (but might be due to use of jargon). Also, what is meant with grid-mean? I have really no idea what global grid-mean is supposed to be.*

A: Yes, the reviewer's understanding is correct. The term 'grid-mean' is commonly used in climate modeling to indicate averaging over both the clear and cloudy part of a grid cell. Given the confusion of the reviewer, we re-considered our terminology and now use instead "all-sky' CRE for the RFO-weighted CRE of a CVS class to distinguish it from the cloudy column or overcast CRE. This "all-sky" terminology conveys the fact that subcolumns with no cloud or a CVS of another class have zero contribution to the CRE of the CVS class of interest when RFO weighting is used. Furthermore, "Cloudy-column" (CVs-specific) CRE is now referred to as "overcast" CRE throughout the text, and its mean represents the average CRE of a CVS class when it occurs.

*page 12: I understand that f and r are expressions for parameters that have been used earlier in the manuscript. Why not introduce them at the first instance they are being used? It would be much less confusing if you would be consistent with the naming parameters throughout the text!*

A: Thank you for the suggestion which we have implemented.

*Please make use of the sub-figure labelling (a, b, c, etc.) in the discussion of the findings. This would make it much easier for the reader to follow your argument.*

A: The text has been revised to include panel labelling.

*page 13, lines 23-28: This is a somewhat disappointing conclusion. What's the contribution of this study apart from anything's possible?*

A: It may be a somewhat disappointing conclusion, but the existence of compensating errors in GCMs is a well-known fact that is confirmed yet again and documented with substantial detail in this paper. Nevertheless, our results contain specific findings that point to specific clouds processes that need to be fixed in order to improve the overall performance of the model.

*page 20, line 19: please update reference.*

A: Done, thank you.

*Figure 1: I took notice of the dotted lines in the figure only after trying to make sense of the combination of classes from your description myself. The purpose of those lines is not stated anywhere. Please add: dotted lines show which cloud classes have been combined for this study.*

A: Thank you again, we added this in the figure 1 caption.

*Figures 2 and 3: I would suggest to revise these figures in a more intuitive way. For instance, you could have columns with high clouds on top and low clouds at the bottom. Also have all cases in Figure 2a in one column, then the plots of Figure 2b next to it in another column, and then the findings of Figure 3 in a third column. However, Figure 3 seems pointless to me as it presents the*

*differences between the two sub-column generators. It would be better to present the difference to the observations for each sub-column generator, i.e. as an individual column added to Figure 2. I would also prefer to see plots from -180 to +180 degree. Finally, please elaborate on the global mean values. Shouldn't they add up to 100%. They don't right now.*

A: First of all, we like maps from 0 to 360 (centered at the dateline) better because they accentuate tropical deep convection around the ITCZ. Second, both observed and modeled CVS RFOs (including clear skies) add up to 100% within rounding error. Regarding the re-organization of the panels, we prefer to keep it the way it is, i.e., moving from clear to single layers CVS, to two-layer CVS, and finally to the three-layer CVS (simple to more complex). Subtracting two model results (Fig. 3) allows us to obtain more spatially coherent results (since the various CVS classes tend to occur in the same regions) and to use a more limited range of differences for the colorbar, thus amplifying the difference maps. If we were to subtract each model configuration from observations, we would need double the number of panels.

*Figures 4 to 6: Please add a line at zero. See also my comments regarding the discussion on page 11. In Figure 6, you should at a statement to the caption regarding the meaning of the grey bars.*

A: Good suggestion to add a zero line, which we have implemented. All bars in Fig. 6, including the gray bar have been defined by the legend and are now defined again in the expanded caption.

*Figures 7 to 9: I'd suggest to refer to the two lines as observations and model.*

A: Done.

*Anonymous Referee #2*

*The simulated cloud radiative effects (CRE) are commonly biased in most climate models. This study, with the aid of both CloudSat/CAPLIPSO retrievals and the implemented subcolumn cloud generator in NASA's GEOS-5, explores the CRE biases in association with cloud vertical structure classification. Results show that while the simulation of global CRE is in much agreement with observation, the CRE due to different cloud classification is not. Moreover, by decomposing model's CRE errors into components stemming from biases in RFO and cloudy-column CRE, the relatively good simulations of global grid-mean CRE largely benefit from compensating errors in these two terms. The method introduced in this paper can be used in other models to explore their CRE behavior, thus beneficial for the modeling community.*

*While I generally find the manuscript suitable for publication in GMD, further improvements are needed before the manuscript is accepted. Below I have included a list of the major comments that I think should be addressed, followed by a list of more specific comments.*

*Major comments:*
*1. The study uses 2B-FLXHR-LIDAR product as a guiding reference for CRE, which was obtained by invoking radiative transfer algorithm operating on thermodynamical fields from re-analysis and cloud properties from CloudSat/CAPLIPSO retrievals. As the authors mentioned, the SW CREs in 2B-FLXHR-LIDAR is strongly time dependent. It is worth to add CERES data as a reference as well, since a great many models are commonly tuned to resemble CERES observations.*

A: The CVS class occurrences and associated radiative fluxes from 2B-FLXHR-LIDAR (obtained in the way that the reviewer describes) are defined at the scale of individual CloudSat profiles, i.e., at approximately 2 km horizontal resolution. Producing the observed global/zonal average results in our study still requires this flux-CVS type mapping at the CloudSat profile horizontal scale. That is the main reason we cannot use CERES: even at the footprint level, CERES radiative fluxes are coarser than the scale at which a CVS occurrence is defined, so such a mapping cannot be achieved. In the model, it is the subcolumn generator that allows us to define CVS occurrences within a grid cell.

*2. Besides cloud overlap assumptions, the cloudiness vertical profile per se is important for the determination of CVS classification. The authors are suggested to replenish the role of layer cloud fraction when revising the paper. The CVS classification may suffer from poor representation of subgrid cloud condensation and/or overlap assumption. In addition, the vertical resolution in GCMs is typically coarser than that in CloudSat/CALIPSO retrievals, which to some extent plays an important role in calculating RFO. The authors need point out this in the paper.*

A: The CVS classification of Oreopoulos et al. (2017) used in this paper does not depend on cloud fraction overlap because layer cloud fractions are either zero or one in the 2B-CLDCLASS-LIDAR profiles. In other words, we are considering only cloud *occurrence* overlap at different vertical layers to assign cloud profiles to a CVS class. We just wanted to clarify this point once again. But we believe that what the reviewer refers to here is the quality of the model cloud profiles passed to the subcolumn generator. These may indeed affect our results. So, what we are really evaluating here is both the quality of the model's cloud scheme in terms of mean profiles, but also the quality of the subcolumn generator to translate these mean profiles into subgrid profiles. The latter issue is partially addressed in this paper by changing the overlap assumption in the subcolumn generator from generalized to maximum-random. One can of course come with other options to create the subgrid columns, and even for generalized overlap, the decorrelation length is a tunable parameter. But both versions of the generator receive as input the same mean profiles, which may be problematic. There is flexibility in changing the generator, but testing different model cloud ("moist") schemes is a bigger effort. Good point about different vertical resolutions. We have added a sentence in the discussion of the concluding section to make the point that "garbage-in" (deficient mean cloud profiles) – "garbage out" (the subcolumn generator will not rectify faulty input profiles).

*3. When comparing the two overlap assumptions, the GN assumption yields more clouds than MR in almost all cloud regimes except for isolated low clouds in extratropics. Why does this occur? Is this because clouds in this region are nonadjacent separated by clear skies that have a vertical distance smaller than specified decorrelation length in GN?*

A: What the reviewer suggests is certainly possible and makes sense. As shown in Oreopoulos et al. (2017), when multiple distinct cloud layers exist in a standard layer, these are still treated as a single entity in the CVS classification. So, the MR scheme sees them as separate layers (random overlap), while the generalized scheme gives a combined fraction less than random because the separation distance is small.

*Specific comments:*
*1. The abbreviation "RFO" is not fully spelled at the first place in the main text.*

A: We fixed this, thanks.

*2. P9L1 regions pronounced orography -> regions with pronounced orography*
*P9L17 exceeds -> exceed 4. P27 As Fig. 3-> As Fig. 4.*

A: Corrections applied, thanks.

**An evaluation of clouds and radiation in a Large-Scale Atmospheric Model using a Cloud Vertical Structure classification**

by

Dongmin Lee[1,2], Lazaros Oreopoulos[2], and Nayeong Cho[3,2]

1. Morgan State University

2. NASA Goddard Space Flight Center

3. University Space Research Association

Confidential manuscript revised for the

*Geoscientific Model Development*

**Corresponding author address:**

*Dongmin Lee*

*NASA-GSFC*

*Code 613*

*Greenbelt MD 20771*

*Dongmin.Lee@nasa.gov*

**Abstract**

We revisit the concept of Cloud Vertical Structure (CVS) classes we have previously employed to classify the planet's cloudiness (Oreopoulos et al., 2017). The CVS classification reflects simple combinations of simultaneous cloud occurrence in the three standard layers traditionally used to separate low, middle, and high clouds and was applied to a dataset derived from active lidar and cloud radar observations. This classification is now introduced in an Atmospheric Global Climate Model, specifically a version of NASA's GEOS-5, in order to evaluate the realism of its cloudiness and of the radiative effects associated with the various CVS classes. Such classes can be defined in GEOS-5 thanks to a subcolumn cloud generator paired with the model's radiative transfer algorithm, and their associated radiative effects can be evaluated against observations. We find that the model produces 50% more clear skies than observations in relative terms, and produces isolated high clouds that are slightly less frequent than in observations, but optically thicker, yielding excessive planetary and surface cooling. Low clouds are also brighter than in observations, but underestimates of the frequency of occurrence (by ~20% in relative terms) help restore radiative agreement with observations. Overall the model reproduces better the longwave radiative effects of the various CVS classes because cloud vertical location is substantially constrained in the CVS framework.

[revised manuscript text omitted]